# The Effect of Sheep and Cow Milk Supplementation of a Low Calcium Diet on the Distribution of Macro and Trace Minerals in the Organs of Weanling Rats

**DOI:** 10.3390/nu12030594

**Published:** 2020-02-25

**Authors:** Keegan Burrow, Wayne Young, Michelle McConnell, Alan Carne, David Barr, Malcolm Reid, Alaa El-Din Bekhit

**Affiliations:** 1Department of Food Science, University of Otago PO Box 56, Dunedin 9054, New Zealand; 2Department of Wine, Food and Molecular Biosciences, Lincoln University, PO Box 85084, Lincoln 7647, New Zealand; 3AgResearch Ltd, Grasslands Research Centre, Private Bag 11008, Manawatu Mail Centre, Palmerston North 4442, New Zealand; Wayne.Young@agresearch.co.nz; 4Riddet Institute, Massey University, Massey University, Private Bag 11222, Palmerston North 4442, New Zealand; 5Department of Microbiology and Immunology, University of Otago PO Box 56, Dunedin 9054, New Zealand; michelle.mcconnell@otago.ac.nz; 6Department of Biochemistry, University of Otago PO Box 56, Dunedin 9054, New Zealand; alan.carne@otago.ac.nz; 7Department of Chemistry, University of Otago PO Box 56, Dunedin 9054, New Zealand; dbarr@chemistry.otago.ac.nz (D.B.); malcolm.reid@otago.ac.nz (M.R.)

**Keywords:** milk, macro-minerals, trace-minerals, ICP-MS, sheep milk, Cow milk, Calcium, Phosphorous, Soft organs

## Abstract

The aim of this study was to investigate the effect of either sheep or cow milk supplementation to a low calcium and phosphorus diet on growth and organ mineral distribution in weanling rats. Rats were fed diets consisting of either a control chow, a 50% reduced calcium and phosphorous chow (low Ca/P), low Ca/P and sheep milk, or low Ca/P and cow milk diet for 28 days. Food intake of the rats, the growth rate of the rats, and the concentrations of minerals in the soft organs and serum were determined. Rats fed the low Ca/P diet alone had lower weight gain than rats consuming either of the milk-supplemented diets (*p* < 0.05). Both sheep milk and cow milk supplementation overcame the effects of consuming a diet restricted in calcium and phosphorus but the sheep milk was effective at a significantly lower level of milk intake (*p* < 0.05). Significant differences (*p* < 0.05) in essential and trace mineral concentrations due to milk type were observed in the kidney, spleen, and liver. For non-essential minerals, significant differences (*p* < 0.05), related to diet, were observed in all organs for arsenic, cesium, rubidium, and strontium concentrations.

## 1. Introduction

Mineral deficiencies during youth and adolescence (up to the age of 25) impact the proper development of various anatomical systems (bones, soft organs etc.) [1,2]. In addition, chronic low-level mineral deficiencies result in short term symptoms (lethargy, nausea, and weakness) [3,4]. As identified by Gharibzahedi, and Jafari [5], the most commonly reported effects result from a lack of adequate iron (Fe), zinc (Zn), iodine (I), and selenium (Se). The diagnosis of deficiency is often related to short term symptoms and ease of testing, which may not reflect the true rate in the population [5]. For example, Fe deficiency has been identified at rates of up to 40% in preschool children, but methods for Fe deficiency testing vary, ranging from the visual hemoglobin color scale to the cyanmethemoglobin analytical colorimetric method [2]. With respect to calcium (Ca), the occurrence of clinical deficiency is extremely rare apart from in extreme dietary circumstances [6]. The long-term impact of low level Ca deficiency during critical growth phases has an impact on growth rates, especially in the context of skeletal development, often leading to a restriction in bone growth [6].

The potential for mineral(s) deficiency leading to health complications in an individual can be related to a range of factors, including age, illness, allergy, genetic predisposition, and diet [7]. Although it is often possible (and sometimes necessary in acute cases) for diet supplements to be used in the short to medium term, long-term diet supplementation is often seen as an ineffective treatment method. This is because mineral and/or nutritional dietary supplements are expensive, inefficient, can cause secondary health effects, and have the risk of over-supplementation [8]. In comparison to long term diet supplementation, the use of diet augmentation of food products is recommended [9].

Mineral deficiency is affected not only by total mineral intake but also mineral absorption rates [5]. Milk has been shown to have some positive effects on mineral absorption, independent of the high mineral concentration present [6]. However, effects are likely to be different between milk from different species due to differences in composition [10]. Specifically, cow milk (CM), buffalo milk, and goat milk may have beneficial effects on Ca absorption [11]. Ca and phosphorus (P) are physiologically critical minerals, which are specifically required for growth and energy production [6]. It is well established that dairy products generally play a key role in providing dietary Ca and P, typically through the consumption of CM, or CM derived products.

The trace and non-essential mineral profiles of sheep milk (SM) have had limited investigation [12]. The concentration of selected toxic minerals of SM has been studied by Yabrir et al. [13] and Ivanova et al. [14]. These studies identified that the concentration of these minerals is highly dependent on the location of milk production (i.e., environmental factors) and specific farming practices [15]. We have previously provided some insight into the effects of New Zealand SM consumption in addition to a diet containing a balanced mineral profile [16,17]. Despite SM having a higher mineral concentration and macronutrients (protein and fat content) compared to CM, its consumption did not have any effect on the development of rats fed a balanced diet [16,17]. The livers of rats that consumed SM had a lower Fe content compared to CM fed animals. Higher concentrations of rubidium (Rb) and cesium (Cs) were present in the brain, kidney, liver, spleen, and serum of the rats fed SM compared to those fed CM [16]. However, the absorption of all types of minerals (macro, trace, and non-essential minerals) occurs at different rates during dietary deficiency, dietary adequacy, and dietary excess [18]. Therefore, this previous work gives insight into the effects of SM minerals in the context of a balanced diet with excess nutrients/minerals supplied by the milk.

Previous studies have established the usefulness of using rat models fed Ca restricted diets to investigate mineral metabolism and bone structure [19,20]. Rader, et al. [20] used weanling male rats to investigate the effects of a diet restricted in Ca (98% reduction) and in P (93%) on a range of serum biomarkers and reported that parathyroid hormone concentration was increased and vitamin D3 concentration was decreased during an 8-week feeding trial (*p* < 0.05).

There are no reports in the literature on the effect of SM consumption with a mineral-deficient diet. Therefore, this study was designed to investigate the effect of consumption of SM or CM to a diet low in Ca and P on the macro, trace, and non-essential mineral distribution in the organs of weanling rats.

## 2. Materials and Methods

The experiment design and the number of animals used in the feeding trial were determined by a biometrician, to maximize statistical power and minimize the number of animals required. An unbalanced experimental design was chosen after consideration of previous work [16,17]. The AgResearch Grasslands Animal Ethics Committee (Palmerston North, New Zealand) independently reviewed and approved all animal experiments covered in this work (application number 14440) as a requirement under the New Zealand Animal Welfare Act [21].

### 2.1. Animals

Newly weaned (three to four week old) male Sprague Dawley rats were sourced from the Hercus Taieri Resource Unit (University of Otago, Dunedin, New Zealand). Upon arrival at the AgResearch small animal facility (Palmerston North, New Zealand), the rats were weighed, individually housed, fed a standard rodent chow diet (AIN-93M diet (Research Diets, New Brunswick, NJ, USA)), and had access to drinking water ad libitum. After 24 h, the rats were weighed again and randomly allocated to one of four groups (two groups of 9 animals and two groups of 15 animals) ensuring even distribution of the animal weight ranges, to achieve a similar mean starting weight and standard deviation for each group. The rats were fed a standard rodent chow diet (AIN-93M diet) and weighed every second day until seven days later when the animals were transferred onto the experimental diets as described below. The overall mean rat weight at the start of the trial feeding period was 124 ± 22.2 g (mean ± standard deviation).

### 2.2. Diets and Procedures

All animals were provided with either a modified-AIN-93M diet (Control) or a low calcium and phosphorus (Low Ca/P) AIN-93M diet (Low Ca/P control, Low Ca/P + SM, and Low Ca/P + CM) ad libitum. The modified-AIN-93M diet was based on a standard rat basal chow diet modified to contain no dairy proteins. The Low Ca/P (AIN-93M) diet is identical to the modified-AIN-93M diet except that it was formulated with a 50% reduction in both Ca and P as shown in Table 1. Fresh diet pellets were provided to the rats every two to three days and the mass consumed by each rat was recorded. The rats were weighed daily during the feeding period.

The diet groups were as follows: 1) Control (modified-AIN-93M) diet (control, *n* = 9); 2) low calcium and phosphorus control (Low Ca/P control, *n* = 9); 3) low calcium and phosphorus with sheep milk (Low Ca/P + SM, *n* = 15); and 4) low calcium and phosphorus with cow milk (Low Ca/P + CM, *n* = 15).

The milk-fed animals (Low Ca/P + SM and Low Ca/P + CM) were provided with whole un-pasteurized milk twice daily, once in the morning and once in the evening (defrosted from frozen aliquots prior to each feeding time point). The cow and sheep milk were sourced from commercial organic farms in the Manawatu and Hawke’s Bay regions of New Zealand, respectively. The milk used in the trial was separated into individual aliquots prior to freezing from a bulk delivery of unprocessed food grade milk, provided by each respective supplier. The amount of milk offered was more than the expected total consumption per day (based on earlier observations) and adjusted during the trial as necessary [16]. The volume of milk consumed by each animal was recorded and the remaining milk from each session was discarded. The control animals were provided with fresh drinking water ad libitum.

After 28 d of feeding, all rats were euthanized by CO_2_ asphyxiation and cervical dislocation. Blood samples were collected by cardiac puncture using 8.5 mL BD Vacutainer^®^ SST™ tubes (Bristol Circle (Franklin Lakes, USA)) and allowed to clot before centrifuging (Sigma 3–18K Centrifuge, Sigma-Aldrich, St. Louis, MI, USA) at 4 °C at 1500 × g for 10 min to separate the serum fraction. The serum was harvested and subsequently stored frozen at −80 °C. Soft tissues (liver, kidney, brain, and spleen) from the rats were harvested, weighed, and frozen in liquid nitrogen before storage at −80 °C. 

### 2.3. Proximate Analysis of Diets and Milk

The nutritional compositions of the basal diets were reported based on supplier information. The nutritional compositions of the milk samples were determined by the MilkTestNZ™ (Hamilton, New Zealand) standardized CM program using a Milkoscan™ (Foss Milkoscan, Foss, Hillerød, Denmark).

### 2.4. Analysis of Mineral Composition

The mineral composition of the milk types, AIN-93M diets, soft organs and serum were determined by inductively coupled plasma mass spectrometry (ICP-MS) as described by Burrow, et al. [16]. Samples were digested in an ultraclean, metal-free Class 100 (ISO 5) laboratory (Department of Chemistry, University of Otago, Dunedin, New Zealand). Microwave digestion was conducted using a MARS 6 microwave digestion unit (CEM Corporation, Matthews, NC, USA) with Mars X-Press digestion tubes (CEM Corporation, Matthews, NC, USA). The analysis was carried out using an ICP-MS (Agilent 7900, Agilent, Santa Clara, CA, USA) instrument in general-purpose plasma mode and with a quartz 2.4 mm torch. The sample depth was set to 10 mm, with a gas flow rate of 1.05 L/min and a nebulizer flow rate of 0.1 rps. Internal standards of beryllium (Be), scandium (Sc), germanium (Ge), rhodium (Rh), indium (In), terbium (Tb), and bismuth (Bi) were added online. Detection limits are reported in Appendix A.

### 2.5. Statistical Analysis

Analysis of variance (ANOVA) was used to determine the differences between each diet with respect to the mineral profiles using SPSS 24 (IBM Corporation, New York, NY, USA). To identify differences between each diet with respect to weight gain, food intake, and milk intake, repeated measures ANOVA was applied (using SPSS 24). Following ANOVA and repeated measures ANOVA, Tukey’s post hoc testing was used where appropriate.

The effect of diet on the mineral concentration in each organ was assessed with the Kruskal Wallis and the Dunns test with Bonferroni correction (using SPSS 24). Relationships between the intake of each mineral and the concentration of each mineral in each individual organ were established with Spearman’s correlation (using SPSS 24). Data are presented as the mean ± the standard deviation. Significance was determined as *p* < 0.05.

## 3. Results and Discussion

### 3.1. Diet Composition

The nutritional compositions of the diets provided to the rats during the feeding trial are shown in Table 2 and the specific mineral intakes for each diet group are shown in Table 3 (derived from data provided in Appendix A). The basal diet used in this study was modified to remove dairy protein from the formulation and replace this with beef protein, in order to prevent any confounding effects. The protein used in rodent diets was from several sources including egg albumin, goat milk, CM, and beef protein [22,23,24]. With respect to milk composition, the data presented (Table 2) are consistent with the expected nutritional composition for both SM and CM as reported in the literature [25].

The data presented in Table 3 show that there is, for the most part, a clear delineation between the minerals provided via the basal diet and those provided via the different milk types. For example, cerium (Ce), chromium (Cr), erbium (Er), uranium (U), vanadium (V), and yttrium (Y) were all higher in the two control diet groups (Control and Low Ca/P control), whereas, Cs, lanthanum (La), and strontium (Sr) were higher in the milk containing diet groups. Although some work has been conducted previously on SM, a comprehensive trace and non-essential element profile of New Zealand SM has not been published [12,16].

It has been previously noted in the literature that sheep milk consistently contains elevated levels of non-essential minerals in comparison to the most common ruminant milk type, cow milk [10,12]. Of particular interest are the concentrations of Al, Cu, and Pb, because these elements are known to have a negative impact on human health. For example, excessive exposure to Pb has been well linked to neurodegeneration [26]. The key mechanisms behind the elevated concentration of these elements has not been established [10,12]. One aspect to note is the overall elevated concentration of all nutritional components in sheep milk. This means that although there can be elevated overall concentrations of individual non-essential minerals when sheep milk is compared to cow milk, the concentration of each mineral is proportionally similar between the two milk types [16]. When disproportionately higher concentrations of non-essential minerals in sheep milk (when compared to cow milk) have been noted, this is typically related to on farm variables, including animal feeding patterns and contaminated milking equipment [10,27]. In the context of the data presented in the present work (Table 3 and Appendix A) the levels observed are not of concern and are within the legal limits for New Zealand and Australia [28].

### 3.2. Rat Food and Milk Intake

The food and milk intakes for each diet group are shown in Figure 1. Milk intake (g/g body weight/day) by the Low Ca/P + CM fed rats was significantly higher than for the Low Ca/P + SM fed rats (*p* < 0.05, Figure 1). The Low Ca/P diet did not result in a significantly different total food intake (g/g body weight/day) by the rats compared to the control diet (*p* > 0.05, Figure 1). The consumption of both kinds of milk significantly reduced the food intake compared to the Low Ca/P control diet (*p* < 0.05, Figure 1). Only the Low Ca/P + SM diet fed rats had a significantly lower food intake than the Control fed rats (*p* < 0.05, Figure 1).

The lower food intake for the Low Ca/P + SM rats compared to the Control fed rats (*p* < 0.05) is most likely due to the high energy density of SM [10]. Larue-Achagiotis et al. [29] showed that adult male Wistar rats that were given the opportunity to self-select dietary components ab libitum, did not have significantly different calorific intakes when compared with rats fed solely on a standard ‘balanced’ basal diet (*p* > 0.05). A self-regulated dietary intake could explain the difference in diet intake between the Low Ca/P + SM fed rats and the Control fed rats as well as most likely being the justification for the lower intake of milk that was observed in the Low Ca/P + SM fed rats compared to the Low Ca/P + CM fed rats (*p* < 0.05, Figure 1).

### 3.3. Rat Weight Gain

The weight gain for each diet group is presented in Figure 2. The Low Ca/P control rats had a significantly (*p* < 0.05) lower weight gain compared with rats that consumed either of the milk diets (Low Ca/P + CM or Low Ca/P + SM). No significant difference in weight gain was observed between the Control diet and the Low Ca/P control fed rats.

Lobo et al. [19] reported body weight gains of 5.61 and 4.72 g/day for control (diet contained 5.51 g Ca/kg) and Ca restricted (diet contained 1.57 g Ca/kg) 4 week old male Wistar rats, respectively, over the course of 33 days. Similar to the present study, the difference in weight gain reported by Lobo et al. [19] was not significantly different between the control and Ca restricted groups (*p* > 0.05). The consumption of either SM or CM with a Low Ca/P diet significantly increased the weight gain of rats (*p* < 0.05, Figure 2), despite the rats consuming either milk type consuming significantly lower amounts of the basal diet compared to the Low Ca/P control fed rats (*p* < 0.05, Figure 1). Although the reduced food intake might be due to the additional energy that the SM and CM provides to the rats, the present study indicates that the change in weight gain is not driven by excess nutrition. This is due to the Low Ca/P + SM fed rats consuming significantly less basal food than the Control fed rats (*p* < 0.05, Figure 1), but had the same weight gain (*p* > 0.05, Figure 2), showing that either SM or CM as a dietary supplement can prevent the reduced weight gain associated with a Ca and P restricted diet.

The restriction of specific minerals, for example altering the Zn concentration in the diet, has been shown to alter feeding patterns in rats. Reeves [30] reported that a reduction of Zn content in the basal diet by 30 fold (from 30 to < 1.0 mg Zn/kg) for 3 days resulted in a significant reduction in both the food intake and weight gain in male Sprague-Dawley rats (*p* < 0.05). It must be noted that Zn specifically has been linked to the regulation of neuropeptide Y, a pancreatic polypeptide associated with hunger [31]. No association has been identified relating the deficiency of Ca or P to food intake in humans.

It must be noted that the reduction in Ca content in the food used in the present study was not as extreme as that reported by other authors in the literature. Previous studies that reported the use of a Ca reduced diet in rats applied reductions of 90% or more of Ca in the basal feed with only one mineral altered [20,32]. In the present work, a lower level of reduction was selected based on ethical consideration. In addition, the selection of a combined reduction in both Ca and P was selected due to the interrelationship that has been identified between the two minerals within both the milk matrix and mineral metabolism [10,33]. The significant differences in weight gain between the Low Ca/P control rats and that of either of the milk diets (Low Ca/P + CM or Low Ca/P + SM) (*p* < 0.05, Figure 2), shows that this study was able to successfully balance animal welfare and reach significant physiological effect that could support the study objectives.

### 3.4. Macro and Trace Minerals in Soft Organs and Serum

The macro and trace mineral composition of rat organs and serum is shown in Table 4 and Table 5. Minerals present below detection limit are not reported. Significant differences in mineral concentrations due to diet were observed in the kidney, spleen, and liver (*p* < 0.05), but not in the serum or brain (*p* > 0.05, Table 4 and Table 5). Correlations between the daily intake of minerals and the organ concentrations of macro and trace minerals are reported in Appendix A.

The macro and trace mineral accumulations in the organs were not different in rats that were fed either control or reduced Ca and P intake diets (*p* > 0.05, Table 4). Significant differences in the Fe, Mo, and Na concentrations in spleen were found between the milk diet fed rats and the Low Ca/P control fed rats (*p* < 0.05), but these three minerals were not significantly different when the milk diet fed and the Control fed rats were compared (*p* > 0.05). This is most likely due to the overall homeostatic regulation of minerals, which is reported to be affected only in the case of extreme deficiency (or excess) [34].

Co concentration was reduced in the spleen and liver related to supplementation with either CM or SM (Table 4, and Appendix A), and had a significant negative correlation with Ca intake in the spleen (*p* < 0.05). Although Co shares absorption pathways with Fe and Ca, the functional role of Co in biological pathways is not fully characterized [35]. The occurrence of Co deficiency is not common in humans, with no consistent data on its prevalence [35,36]. Co is a cofactor for the enzymes methionine synthase and methylmalonyl-CoA mutase in the biologically active form cobalamin [36]. Co has the lowest recommended dietary intake of all trace elements at 2.4 μg per day in humans [37]. However, the supplementation of the diet of sheep with Co is quite common, because the absorption of Co by sheep is extremely ineffective [38]. This may explain why the SM containing diet in the present study has such an elevated level of Co. For rats, no target Co intake level has been reported [35]. Therefore, it can be concluded that the Co concentration observed in the present study is not detrimental.

Significant negative correlations between the intakes and organ concentrations were observed for Mn (liver) and Mo (spleen and liver) (*p* < 0.05, Appendix A). Although Mn absorption is poorly characterized, some evidence exists for interactions with Fe during absorption [39]. Hansen et al. [40] identified that Mn accumulation in the liver is significantly reduced in pigs that were fed a control diet that was supplemented with Fe (at 500 mg Fe/kg of diet), compared to feeding a control diet (*p* < 0.05). Like Mn, Mo absorption is similarly poorly characterized. Although one Mo specific transport protein has been reported in mammals, the pathway is poorly understood [41].

A relationship between Zn intake and liver Zn concentration is reported in Appendix A, where there was a significant negative correlation between Zn intake and liver Zn concentrations (*p* < 0.05). Even though the rats consuming either of the control diets (Control or Low Ca/P control) had the highest Zn intakes (Table 3), they also had the lowest liver Zn content (Table 4). The relationship between Ca and Zn accumulation in organs and consumption of dairy products has not been fully established. Hansen et al. [42] found that the addition of Ca and/or casein phosphopeptides to bread fortified with Zn did not have a significant effect on apparent Zn absorption (measured as a function of whole body ^65^Zn retention) in healthy human adults (aged 19 to 30 years). In contrast, Miller et al. [43] used the data from 72 studies to devise a mathematical model of Zn absorption that showed both Ca intake and protein intake was positively associated with Zn absorption, whereas phytate intake was strongly negatively associated. A high rate of Zn deficiency has been found worldwide with a deficiency risk of 16 ± 1.4% [44]. As discussed above, even though the rats that consumed either of the control diets had the highest Zn intakes (Table 3), they had the lowest Zn concentrations in the liver (Table 4). It appears that the consumption of SM as a diet supplement may be able to increase Zn absorption; however, the data do not provide any indication of the mechanism(s) involved in this interaction. Therefore, further targeted work focusing on the relationship between SM intake and Zn absorption (rather than accumulation) is required.

Cu intake had a significant negative correlation with organ concentration in both the liver and kidney (*p* < 0.05, Appendix A). Antagonistic interactions between Cu and Fe absorption have been reported in the literature in human studies [45]. Furthermore, higher dietary Zn concentrations (rather than absorbed levels of Zn) have been shown to block Cu absorption through competitive transport interactions [46]. Rats fed the Low Ca/P diet plus SM had a significantly higher Cu concentration in the organs higher than the levels of the Control diet (*p* < 0.05, Table 4), but not the diet plus CM (*p* > 0.05, Table 4). It is, therefore, inconclusive as to whether SM specifically, or milk generally, can improve Cu absorption. Literature reports indicate that milk type can affect the Cu bioavailability. Díaz-Castro et al. [47] showed that goat milk (fortified with Ca) significantly increased the bioavailability of Cu compared with CM (fortified with Ca) in male Wistar albino rats with induced iron deficient anemia (*p* < 0.001). Targeted work focusing on the relationship between SM intake and Cu absorption (rather than accumulation) needs to be undertaken to understand the process in this case.

The essential and trace mineral concentrations in the organs and serum observed in this study were within the reported range for growing rats [48,49]. Most of the differences observed with respect to the essential and trace mineral concentrations occurred in spleen and liver (Table 4), which relates to the role these organs have in mineral metabolism, regulation, and storage [50].

### 3.5. Non-Essential Minerals in Soft Organs and Serum

Significant differences in concentration for a variety of non-essential minerals due to diet were observed in all rat organs for As, Cs, Rb, and Sr (*p* < 0.05, Table 4 and Table 5). Correlations between the daily intake and organ concentrations of non-essential minerals are reported in Appendix A. Minerals present below detection limits (Appendix A) are not reported.

As was found at concentrations above the detection limit in the soft organs of the rats (Table 4). In the brain, rats fed on the Low Ca/P + CM diet showed significantly lower As concentrations than rats fed either control diet (Control and Low Ca/P control) (*p* < 0.05, Table 4). In the kidney, the Low Ca/P + SM showed a significantly lower As concentration than that of the Low Ca/P control fed rats (*p* < 0.05, Table 4). In the liver, the Low Ca/P + CM fed rats had the lowest As concentration, which was significantly lower than that of the Low Ca/P control fed rats (*p* < 0.05, Table 4). In the spleen, with respect to As, there was no statistical overlap between the control diets (Control or Low Ca/P control) and the milk diets. The control diets resulted in significantly lower As concentrations in the spleen than the milk diets (*p* < 0.05, Table 4). It must be noted that despite As not being found in the various dietary components above the detection limit (Appendix A), accumulation of As still occurred in all soft organs tested (Table 4). The concentrations of As found in the present study are within the expected range for rat organs. Shimamura, et al. [48] showed that 17-week-old female Wistar rats had liver As concentrations of 570 ± 180 µg/kg. For kidney, Shimamura, et al. [49] showed that 17 week old female Wistar rats had an average As concentration of 810 ± 200 µg/kg. When our data are compared with this published literature, it indicates that although As accumulation did occur, the accumulation was not of concern for animal health due to the low concentrations observed.

In the present study, the concentrations of Cs and Rb in the soft organs and serum (Table 4) were consistent with those observed in our earlier study [16]. Cs showed a constant pattern across all four organs (brain, liver, kidney, and spleen) where the control diet (Control and Low Ca/P control) groups and the Low Ca/P + CM fed rats were significantly lower than the Low Ca/P + SM fed rats (*p* < 0.05, Table 4). For Rb in the brain, liver, and kidney, rats fed the control diets (Control and Low Ca/P control) showed significantly lower concentrations compared to the milk diet groups (Low Ca/P +SM, and Low Ca/P + CM) (*p* < 0.05, Table 4). The pattern for Rb was different in the spleen, where the Control diet fed rats showed the lowest concentration, significantly lower than the milk diet fed rats (*p* < 0.05, Table 4), but the Low Ca/P control diet fed rats did not show a significant difference in Rb concentrations compared with the Low Ca/P + CM feed rats (*p* > 0.05, Table 4). Data in Appendix A show a significant positive correlation (*p* < 0.05) between the intakes of Cs and Rb and the concentration in each organ. For Cs and Rb, it is known that these minerals and their accumulation are related to dietary intake [51,52]. Little information is available on the physiological impact of the stable isotopes of Cs and Rb on rat organs [51,52]. As established by Burrow, et al. [16], data currently available for Cs focuses on the effects of the radioactive isotopes ^134^Cs and ^137^Cs in the context of nuclear contamination rather than ‘natural’ environmental levels [53].

Sr had significantly different distributions among the different diet groups in all the tested organs (*p* < 0.05, Table 4). Sr has been shown to have a number of biochemical similarities to Ca [54,55]. The increase in Sr accumulation in the organs of the Low Ca/P control fed rats, therefore, might be due to Sr uptake instead of Ca. The patterns observed for the concentrations of Sr in organs for the Control, Low Ca/P +SM, and Low Ca/P + CM fed rats are closely related to the intakes of Sr. However, the concentration of Sr in the Low Ca/P control fed rats does not follow this pattern. Sr is considered to be a non-essential mineral because it is not known to be required for any biological function, [54,55] although some evidence exists that Sr can play a positive role in bone metabolism by substituting Ca in the hydroxyapatite (HA) structure [54]. A study by Pei et al. [56] using five month old male Sprague-Dawley rats with chemotherapy-induced osteoporosis, demonstrated that the administration of strontium ranelate (at 900 mg/kg/day) resulted in significantly increased bone parameters including trabecular bone volume and trabecular thickness, in comparison to a control treatment. However, further work is still required to establish a specific biological role before Sr can be considered a trace mineral; this is also because the investigations into the role of Sr reported in the literature have focused on bone rather than the role of Sr in other organs [53,55]. In the present study, it appears that the mechanism of Sr absorption is not altered due to the consumption of either SM or CM.

## 4. Conclusions

Based on the data obtained in this current study, the use of New Zealand SM or CM can overcome the negative effects of a Ca and P restricted/deficient diet. It was found that SM was able to prevent the effects of consuming a diet restricted in Ca and P equally compared to CM, but the effect was achieved at a much lower intake of SM. Although significant differences in organ distribution were observed with respect to selected macro and trace minerals related to milk type (specifically Zn, Cu, Co, and Fe) the data generated in the current study does not indicate a unique interaction between SM consumption and mineral absorption, for either macro or trace minerals. With respect to non-essential mineral accumulation, this was found to be intricately linked to the intake of each respective mineral.

## Figures and Tables

**Figure 1 nutrients-12-00594-f001:**
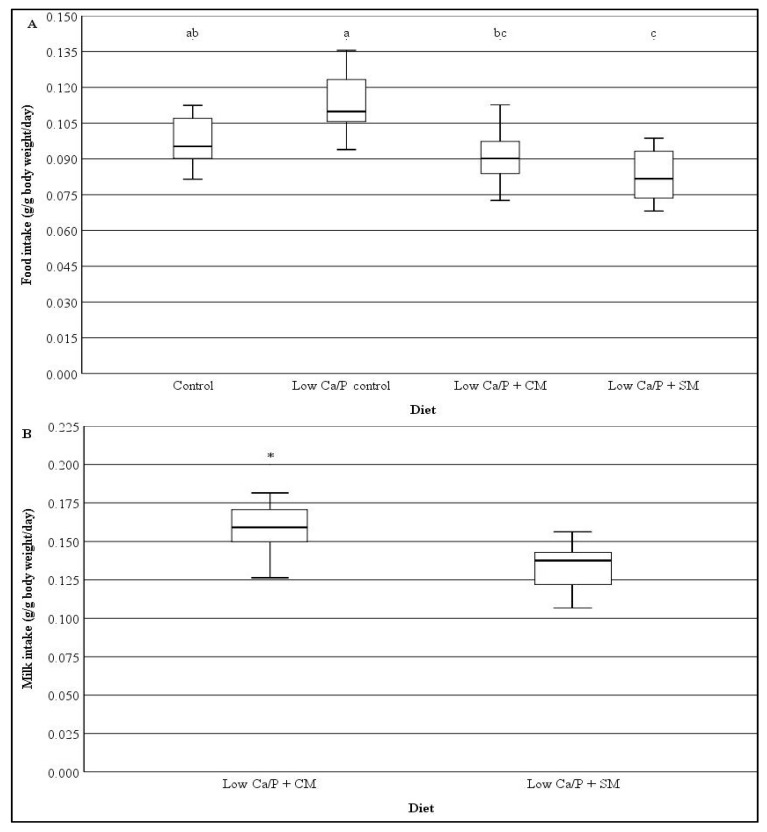
Diet intake of rats fed on different milk diets (mean ± standard deviation): (A) Food intake (g/g body weight/day) and (B) Milk intake (g/g body weight/day). Control = rats fed a modified-AIN-93M diet (*n* = 9), Low Ca/P control = rats fed a Low Ca/P Modified-AIN-39M diet (*n* = 9), Low Ca/P + CM = rats fed a modified-AIN-93M diet with the addition of cow milk ad libitum instead of drinking water (*n* = 15), Low Ca/P + SM = rats fed a Low Ca/P modified-AIN-93M diet with the addition of sheep milk ad libitum instead of drinking water (*n* = 15). Different superscript letters indicate significant differences between dietary groups with respect to food intake, determined using repeated measures ANOVA and Tukey’s post hoc testing (*p* < 0.001). * indicates a significant difference between the different diets with respect to milk intake, determined using repeated measures ANOVA (*p* = 0.001).

**Figure 2 nutrients-12-00594-f002:**
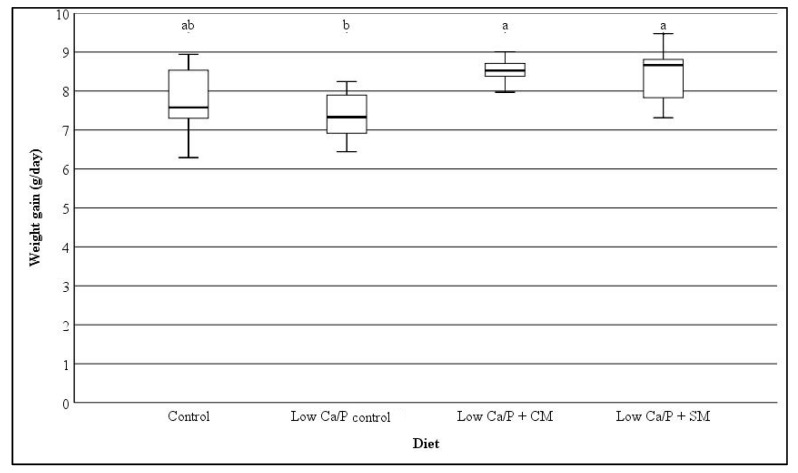
Weight gain (g/day) of rats fed on different milk diets (mean ± standard deviation). Control = rats fed a modified-AIN-93M diet (*n* = 9), Low Ca/P control = rats fed a Low Ca/P Modified-AIN-39M diet (*n* = 9), Low Ca/P + CM = rats fed a Low Ca/P modified-AIN-93M diet with the addition of cow milk ad libitum instead of drinking water (*n* = 15), Low Ca/P + SM = rats fed a Low Ca/P modified-AIN-93M diet with the addition of sheep milk ad libitum instead of drinking water (*n* = 15). Different superscript letters indicate significant differences between dietary groups for animal weight gain, determined using repeated measures ANOVA and Tukey’s post hoc testing (*p* = 0.008).

**Table 1 nutrients-12-00594-t001:** Composition of the modified-AIN-93M and low Ca/P modified-AIN-93M diets used in the animal feeding trial ^*^.

Diet	Modified-AIN-93M (g)	Low Ca/P modified-AIN-93M (g)
Ingredient		
Beef protein extract	140	140
L-cystine	1.8	1.8
Corn starch	495.69	495.69
Maltodextrin	125	125
Sucrose	106.69	106.69
Cellulose	50	50
Soybean oil	40	40
t-butylhydroquinone	0.008	0.008
Mineral mix ^a^	3.5	3.5
NaCl	2.59	2.59
CaCO_3_ ^b^	12.495	6.248
KH_2_PO_4_ ^c^	8.75	4.375
K_3_C_6_H_5_O ^d^	0.98	4.375
Vitamin mix ^e^	10	10
Choline bitartrate	2.5	2.5
Yellow dye ^f^	0.00	0.05

* Values as provided by the supplier. ^a^ Proprietary mix (S10022C). ^b^ 40% Ca. ^c^ 22.8% P and 28.7% K. ^d^ 36.2% K. ^e^ Proprietary mix (V10037). ^f^ Tartrazine.

**Table 2 nutrients-12-00594-t002:** Nutritional composition of diets.

Fraction	Basal Diet *		Milk ^#^	
	Modified-AIN-93M (%)	Low Ca/P Modified-AIN-93M (%)	Cow Milk (%)	Sheep Milk (%)
Protein	14	14	3.69	5.55
Lipid	4	4	3.63^^^	8.72 ^^^
Carbohydrate	76	76	-	-
Lactose	-	-	4.81	4.47

* As reported by the supplier on a dry weight basis. ^#^ As determined by Milkoscan™ (Foss Milkoscan, Foss, Hillerød, Denmark) measurement on a wet weight basis. ^ As determined by Monjonnier ether extraction method (Association of Official Analytical Chemists, 2005a), on a wet weight basis.

**Table 3 nutrients-12-00594-t003:** Mineral intake per day of rats fed on different milk diets ^^^.

Diet		Control [µg/day]	Low Ca/P Control [µg/day]	Low Ca/P + CM [µg/day]	Low Ca/P + SM [µg/day]
Mineral Type	Mineral Intake				
Macro	Ca *	75.4 ± 7.17 ^b^	53.8 ± 3.75 ^c^	93.3 ± 5.44 ^a^	97.4 ± 11.7 ^a^
	Mg *	14.8 ± 1.40 ^ab^	14.0 ± 0.98 ^b^	15.8 ± 0.73 ^a^	15.4 ± 1.38 ^a^
	K *	180 ± 17.1 ^b^	168 ± 11.7 ^bc^	198 ± 9.01 ^a^	160 ± 13.5 ^c^
	P *	56.1 ± 5.33 ^b^	42.7 ± 2.98 ^c^	76.7 ± 4.61 ^a^	79.5 ± 9.68 ^a^
	Na *	123 ± 11.7 ^a^	113 ± 7.85 ^b^	108 ± 5.89 ^b^	113 ± 9.65 ^b^
Trace	Cu	148 ± 14.1 ^a^	117 ± 8.17 ^b^	97.8 ± 6.80 ^c^	96.9 ± 8.02 ^c^
	Fe	180 ± 17.1 ^a^	168 ± 11.7 ^a^	148 ± 9.30 ^b^	170 ± 14.6 ^a^
	Mn	262 ± 24.9 ^a^	220 ± 15.4 ^b^	182 ± 12.9 ^c^	165 ± 14.3 ^c^
	Mo	3.63 ± 0.35 ^a^	2.66 ± 0.19 ^b^	2.18 ± 0.16 ^c^	1.94 ± 0.17 ^d^
	Ni	14.0 ± 1.33 ^a^	12.3 ± 0.86 ^b^	10.1 ± 0.72 ^c^	9.07 ± 0.79 ^c^
	Zn *	1.03 ± 0.10 ^a^	0.93 ± 0.07 ^b^	0.91 ± 0.06 ^bc^	0.84 ± 0.07 ^c^
Non-essential	Al	220 ± 21.0 ^a^	240 ± 16.8 ^a^	197 ± 14.1 ^b^	234 ± 19.8 ^a^
	Ce ^#^	390 ± 37.1 ^a^	303 ± 21.1 ^b^	249 ± 17.8 ^c^	221 ± 19.5 ^d^
	Cr	36.5 ± 3.47 ^a^	31.0 ± 2.16 ^b^	25.5 ± 1.83 ^c^	22.6 ± 1.99 ^d^
	Cs	0.37 ± 0.04 ^b^	0.35 ± 0.02 ^b^	0.33 ± 0.02 ^b^	1.13 ± 0.17 ^a^
	Er ^#^	32.7 ± 3.11 ^a^	30.4 ± 2.12 ^a^	25.0 ± 1.79 ^b^	22.1 ± 1.95 ^c^
	La	2.77 ± 0.26 ^a^	2.56 ± 0.18 ^b^	2.11 ± 0.15 ^c^	1.87 ± 0.16 ^d^
	Li	2.77 ± 0.26 ^a^	2.56 ± 0.18 ^a^	2.11 ± 0.15 ^b^	2.17 ± 0.18 ^b^
	Nd ^#^	215 ± 20.5 ^a^	185 ± 12.9 ^b^	152 ± 10.9 ^c^	135 ± 11.9 ^d^
	Ni	14.0 ± 1.33 ^a^	12.3 ± 0.86 ^b^	10.1 ± 0.72 ^c^	9.07 ± 0.79 ^c^
	Rb	72.2 ± 6.86 ^c^	67.2 ± 4.69 ^c^	131 ± 8.4 ^a^	110 ± 12.6 ^b^
	Sr	41.9 ± 3.98 ^b^	31.6 ± 2.20 ^c^	46.9 ± 2.38 ^b^	75.1 ± 10.1 ^a^
	U	240 ± 22.9 ^a^	153 ± 10.6 ^b^	125 ± 8.99 ^c^	111 ± 9.81 ^c^
	V	12.0 ± 1.12 ^a^	10.0 ± 0.73 ^b^	9.00 ± 0.62 ^c^	8.00 ± 0.67 ^d^
	Y ^#^	511 ± 48.5 ^a^	453 ± 31.6 ^a^	373 ± 26.7 ^b^	331 ± 29.1 ^c^

^ Results are reported as mean ± standard deviation, Control = rats fed a modified-AIN-93M diet (*n* = 9), Low Ca/P control = rats fed a Low Ca/P modified-AIN-93M diet (*n* = 9), Low Ca/P + CM = rats fed a Low Ca/P modified-AIN-93M diet with the addition of cow milk ad libitum instead of drinking water (*n* = 15), Low Ca/P + SM = rats fed a Low Ca/P modified-AIN-93M diet with the addition of sheep milk ad libitum instead of drinking water (*n* = 15), different superscript letters indicate significant differences in mineral intakes between diets within an organ, calculated using ANOVA (*p* < 0.05). * [mg/day]. ^#^ [ng/day].

**Table 4 nutrients-12-00594-t004:** Macro, trace, and non-essential mineral concentrations in the organs of rats consuming milk diets ^^^.

Diet			Control	Low Ca/P Control	Low Ca/P + CM	Low Ca/P + SM
Organ	Element Type	Element	[µg/kg]	[µg/kg]	[µg/kg]	[µg/kg]
Brain	Macro	Ca ^#^	46.3 ± 3.79	46.9 ± 2.70	46.5 ± 2.92	45.9 ± 3.32
		K ^@^	3.64 ± 0.18	3.76 ± 0.20	3.73 ± 0.16	3.66 ± 0.20
		Mg ^#^	148 ± 4.40	151 ± 5.77	150 ± 5.03	149 ± 5.16
		Na ^#^	1.07 ± 0.04	1.10 ± 0.07	1.11 ± 0.04	1.10 ± 0.05
		P ^@^	2.88 ±0.10	2.93 ± 0.14	2.96 ± 0.14	2.95 ± 0.11
	Trace	Cu ^#^	1.85 ± 0.10	1.83 ± 0.10	1.85 ± 0.10	1.91 ± 0.11
		Fe ^#^	17.9 ± 2.14	17.5 ± 1.79	17.2 ± 2.48	18.2 ± 2.76
		Mn	353 ± 35.2	373 ± 22.9	370 ± 16.5	364 ± 17.5
		Mo	26.2 ± 2.67	25.6 ± 2.39	26.1 ± 1.82	25.1 ± 1.79
		Zn ^#^	11.7 ± 0.46	11.9 ± 0.49	11.9 ± 0.43	11.6 ± 0.52
	Non-essential	As	22.2 ± 4.16 ^a^	20.7 ± 2.84 ^ab^	15.1 ± 2.23 ^c^	16.3 ± 3.12 ^bc^
		Co	3.89 ± 0.42	4.06 ± 0.82	3.41 ± 0.41	3.47 ± 0.44
		Cs	5.20 ± 0.79 ^b^	4.87 ± 0.44 ^b^	4.61 ± 0.48 ^b^	9.38 ± 1.74 ^a^
		Rb ^#^	1.29 ± 0.13 ^b^	1.35 ± 0.17 ^b^	2.09 ± 0.22 ^a^	2.05 ± 0.27 ^a^
		Sr	10.7 ± 2.39 ^b^	29.9 ± 17.6 ^a^	13.8 ± 6.16 ^b^	17.8 ± 8.97 ^ab^
		Pb	BDL	BDL	BDL	BDL
Kidney	Macro	Ca ^#^	69.5 ± 8.41	69.2 ± 4.97	67.9 ± 6.03	69.0 ± 5.25
		K ^@^	2.49 ± 0.38	2.71 ± 0.15	2.54 ± 0.28	2.55 ± 0.19
		Mg ^#^	180 ± 26.2	185 ± 9.28	178 ± 17.3	179 ± 13.2
		Na ^#^	1.42 ± 0.15	1.42 ± 0.07	1.40 ± 0.10	1.46 ± 0.14
		P ^@^	2.61 ± 0.40	2.62 ± 0.11	2.54 ± 0.26	2.54 ± 0.21
	Trace	Cu ^#^	3.88 ± 0.54 ^b^	3.38 ± 0.20 ^b^	5.29 ± 1.13 ^a^	5.34 ± 1.01 ^a^
		Fe ^#^	60.0 ± 10.4	57.7 ± 6.58	59.5 ± 7.94	61.9 ± 10.6
		Mn	750 ± 123	829 ± 68.8	755 ± 105	764 ± 78.6
		Mo	180 ± 25.2	184 ± 15.6	183 ± 23.6	187 ± 16.7
		Zn ^#^	18.0 ± 2.14	18.1 ± 1.98	18.0 ± 2.00	18.5 ± 1.46
	Non-essential	As	75.8 ± 19.1 ^ab^	65.1 ± 6.12 ^a^	56.6 ± 10.7 ^ab^	61.0 ± 14.5 ^b^
		Co	85.4 ± 17.5	83.3 ± 11.4	68.9 ± 9.89	66.1 ± 9.26
		Cs	14.0 ± 2.95 ^b^	13.8 ± 1.62 ^b^	12.0 ± 1.37 ^b^	20.2 ± 2.69 ^a^
		Rb ^#^	2.92 ± 0.51 ^b^	3.54 ± 0.37 ^b^	4.96 ± 0.67 ^a^	4.76 ± 0.66 ^a^
		Sr	24.1 ± 4.70 ^b^	46.7 ± 5.22 ^a^	27.6 ± 5.37 ^b^	39.4 ± 8.14 ^a^
		Pb	51.7 ± 38.8	81.5 ± 48.7	55.2 ± 59.0	105 ± 76.1
Liver	Macro	Ca ^#^	33.7 ± 3.56	32.8 ± 3.17	35.4 ± 1.91	36.4 ± 3.35
		K ^@^	3.34 ± 0.27 ^b^	3.41 ± 0.10 ^b^	3.61 ± 0.14 ^a^	3.55 ± 0.19 ^ab^
		Mg ^#^	201 ± 17.6	206 ± 7.91	209 ± 9.70	209 ± 14.4
		Na ^#^	668 ± 45.0	686 ± 37.9	663 ± 46.4	690 ± 49.6
		P ^@^	2.90 ± 0.21	2.93 ± 0.14	3.00 ± 0.18	3.04 ± 0.22
	Trace	Cu ^#^	3.08 ± 0.34 ^b^	3.12 ± 0.13 ^b^	3.34 ± 0.24 ^ab^	3.41 ± 0.19 ^a^
		Fe ^#^	128 ± 22.8 ^a^	138 ± 16.9 ^a^	89.1 ± 9.79 ^b^	88.8 ± 10.6 ^b^
		Mn	1.41 ± 0.16 ^b^	1.57 ± 0.15 ^ab^	1.72 ± 0.15 ^a^	1.79 ± 0.11 ^a^
		Mo	235 ± 23.9 ^b^	214 ± 29.5 ^b^	297 ± 30.7 ^a^	335 ± 37.9 ^a^
		Zn ^#^	20.8 ± 1.75 ^b^	21.0 ± 1.82 ^b^	24.0 ± 1.97 ^a^	25.0 ± 2.06 ^a^
	Non-essential	As	57.1 ± 13.4 ^ab^	64.3 ± 11.4 ^a^	45.8 ± 11.6 ^b^	47.7 ± 12.4 ^ab^
		Co	15.6 ± 1.70 ^ab^	18.1 ± 2.88 ^a^	12.1 ± 2.81 ^c^	12.3 ± 2.44 ^bc^
		Cs	8.80 ± 1.05 ^b^	8.49 ± 0.91 ^b^	8.23 ± 0.80 ^b^	15.6 ± 1.88 ^a^
		Rb ^#^	4.84 ± 0.42 ^b^	4.98 ± 0.54 ^b^	8.51 ± 0.82 ^a^	8.38 ± 0.92 ^a^
		Sr	11.6 ± 1.56 ^c^	17.0 ± 2.07 ^a^	13.0 ± 1.92 ^bc^	15.1 ± 1.44 ^ab^
		Pb	BDL	BDL	BDL	BDL
Spleen	Macro	Ca ^#^	31.2 ± 5.81	36.6 ± 4.88	34.2 ± 7.04	37.3 ± 3.69
		K ^@^	4.12 ± 0.54	4.54 ± 0.36	4.35 ± 0.53	4.47 ± 0.50
		Mg ^#^	187 ± 25.0	207 ± 17.7	199 ± 24.8	206 ± 22.7
		Na ^#^	490 ± 71.4 ^b^	559 ± 65.0 ^a^	551 ± 41.8 ^ab^	545 ± 52.5 ^ab^
		P ^@^	3.05 ± 0.45	3.39 ± 0.32	3.35 ± 0.15	3.39 ± 0.43
	Trace	Cu ^#^	0.94 ± 0.16	0.98 ± 0.09	0.92 ± 0.19	1.00 ± 0.12
		Fe ^#^	305 ± 55.7 ^ab^	427 ± 99.1 ^a^	238 ± 75.0 ^bc^	205 ± 41.5 ^c^
		Mn	150 ± 25.2	178 ± 20.0	154 ± 28.6	162 ± 18.0
		Mo	53.2 ± 14.9 ^ab^	66.1 ± 11 ^a^	50.4 ± 14.9 ^b^	55.4 ± 12.4 ^ab^
		Zn ^#^	16.2 ± 2.28	18.2 ± 1.94	17.3 ± 1.96	18.0 ± 2.03
	Non-essential	As	171 ± 45.7 ^a^	197 ± 37.7 ^a^	116 ± 31.8 ^b^	115 ± 16.5 ^b^
		Co	8.78 ± 1.10 ^a^	10.1 ± 1.71 ^a^	5.89 ± 1.64 ^b^	5.59 ± 0.85 ^b^
		Cs	8.01 ± 2.11 ^b^	8.22 ± 0.94 ^b^	6.52 ± 1.26 ^b^	12.3 ± 1.57 ^a^
		Rb ^#^	3.15 ± 0.40 ^c^	3.73 ± 0.44 ^bc^	5.05 ± 1.13 ^ab^	5.43 ± 0.80 ^a^
		Sr	10.3 ± 1.42 ^c^	19.5 ± 3.24 ^a^	11.8 ± 2.21 ^b^	17.2 ± 2.92 ^a^
		Pb	BDL	BDL	BDL	BDL

^^^ Results are reported as mean ± standard deviation, Control = rats fed a modified-AIN-93M diet (*n* = 9), Low Ca/P control = rats fed a Low Ca/P modified-AIN-93M diet (*n* = 9), Low Ca/P + CM = rats fed a Low Ca/P modified-AIN-93M diet with the addition of cow milk ad libitum instead of drinking water (*n* = 15), Low Ca/P + SM = rats fed a Low Ca/P modified-AIN-93M diet with the addition of sheep milk ad libitum instead of drinking water (*n* = 15), BDL = below detection limit (Appendix A), different superscript letters indicate significant differences in mineral concentrations between treatments within an organ, calculated using the Kruskal Wallis test and Dunn’s test with Bonferroni correction (*p* < 0.05). ^#^ [mg/kg]. ^@^ [g/kg].

**Table 5 nutrients-12-00594-t005:** Macro, trace, and non-essential mineral concentrations in the serum of rats consuming milk diets ^^^.

Diet		Control	Low Ca/P Control	Low Ca/P + CM	Low Ca/P + SM
Element Type	Element	[µg/mL]	[µg/mL]	[µg/mL]	[µg/mL]
Macro	Ca	142 ± 10.8	140 ± 8.73	141 ± 8.06	138 ± 10.2
	K	331 ± 44.1	284 ± 54.3	306 ± 46.5	281 ± 45.3
	Mg	29.1 ± 6.30	28.6 ± 4.46	26.5 ± 3.72	29.9 ± 7.20
	P	199 ± 18.1	181 ± 12.5	187 ± 19.3	186 ± 9.71
Trace	Cu	1.00 ± 0.12	0.94 ± 0.05	1.00 ± 0.10	1.01 ± 0.09
	Fe	3.41 ± 0.91	3.07 ± 1.37	4.20 ± 1.15	4.11 ± 1.17
	Zn	1.57 ± 0.15	1.55 ± 0.17	1.65 ± 0.25	1.70 ± 0.26
Non-essential	Cs *	0.88 ± 0.23^ab^	0.67 ± 0.12^b^	0.66 ± 0.14^b^	1.00 ± 0.17^a^
	Rb	0.28 ± 0.06^ab^	0.22 ± 0.05^b^	0.37 ± 0.09^a^	0.37 ± 0.09^a^
	Sr	BDL	BDL	BDL	BDL
	Pb	BDL	BDL	BDL	BDL

^^^ Results are reported as mean ± standard deviation, Control = rats fed a modified-AIN-93M diet (*n* = 9), Low Ca/P control = rats fed a Low Ca/P modified-AIN-93M diet (*n* = 9), Low Ca/P + CM = rats fed a Low Ca/P modified-AIN-93M diet with the addition of cow milk ad libitum instead of drinking water (*n* = 15), Low Ca/P + SM = rats fed a Low Ca/P modified-AIN-93M diet with the addition of sheep milk ad libitum instead of drinking water (*n* = 15), BDL = bellow detection limit (Appendix A), different superscript letters indicate significant differences in mineral concentrations between treatments within an organ, calculated using the Kruskal Wallis test and Dunn’s test with Bonferroni correction (*p* < 0.05). * (ng/mL).

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
