# Peer review of "The Effect of Sheep and Cow Milk Supplementation of a Low Calcium Diet on the Distribution of Macro and Trace Minerals in the Organs of Weanling Rats"

_nutrients, 2020, doi:10.3390/nu12030594_

Round 1
Reviewer 1 Report
The manuscript is well written and interesting, however I have some comments (see below) that must be taken into consideration before publication:
Line 160: it would be better to use “nutrients” or “nutritional composition” of the diets instead “macro-composition”
Table 2: please provide information about the total minerals content (Ash content) in diets
Line 174: please explain abbreviation if they appear for the first time (CS, La, Sr)
Table 3. I suggest to separate the minerals into subgroups: macrominerals and trace minerals instead of listing them in alphabetical order; while non-essential elements can be presented as one group (listed alphabetically).
Table 3: the superscript letters indicating the statistical significance should be uniform, if the highest value is marked as "a" (as in the case of Ce) then also in the case of other elements the sign "a" must indicate the highest value (and not as in the case of La)
Tables 4: I think that in this table, even if very extensive, the results for the same minerals should be provided for each organ and serum, because the separation into two tables (Tab. 4 with statistical significant differences) and Table 3S (non- significant data) makes it very difficult to assess the impact of diets. Like Tab.3, I suggest to separate trace elements from macroelements.
Line 333-342: please check carefully the text describing the non-essential elements as I found many mistakes concerning As
Line 353-357: also please check carefully the text describing Rb in different organs, again there are some inaccuracies
Author Response
Reviewer 1:
- Line 160: it would be better to use “nutrients” or “nutritional composition” of the diets instead “macro-composition”
The term “macro-composition” has been replaced with “nutritional composition” at the following locations in the document to read as follows:
- Line 135: “The nutritional compositions of the basal diets were reported based on supplier information.”
- Line 136: “The nutritional compositions of the milk samples were determined by the MilkTestNZ™ (Hamilton, New Zealand) standardized CM program using a Milkoscan™ (Foss Milkoscan, Foss, Hillerød, Denmark).”
- Line 165: “The nutritional compositions of the diets provided to the rats during the feeding trial are shown in…”
- Line 171: ”… consistent with the expected nutritional composition for both SM and CM as reported in the literature [25].”
- Line 173: “ Table 2: Nutritional composition of diets”
- Table 2: please provide information about the total minerals content (Ash content) in diets
The ash content was not measured in this study as the ash content of the diet is not important other than as an indicator of the inorganic material, but for this study, the relevant inorganic material (i.e. minerals) were measured directly by ICP-MS.
- Line 180: please explain abbreviation if they appear for the first time (CS, La, Sr)
The abbreviation for Cesium (Cs) is introduced in the introduction (Line 68). The abbreviations for lanthanum (La), and strontium (Sr) are introduced in L177 and now reads:
- Line 180: “… whereas, Cs, lanthanum (La), and strontium (Sr) were higher in the milk containing diet groups.”
- Table 3. I suggest to separate the minerals into subgroups: macrominerals and trace minerals instead of listing them in alphabetical order; while non-essential elements can be presented as one group (listed alphabetically).
Table 3 has been rearranged in order to separate the macrominerals and trace minerals from each other according to the Reviewer’s suggestion
- Table 3: the superscript letters indicating the statistical significance should be uniform, if the highest value is marked as "a" (as in the case of Ce) then also in the case of other elements the sign "a" must indicate the highest value (and not as in the case of La)
The superscript letters in all tables have been double checked. The superscript designations for La have been corrected as follows:
|
La |
2.77 ± 0.26a |
2.56 ± 0.18b |
2.11 ± 0.15c |
1.87 ± 0.16d |
- Tables 4: I think that in this table, even if very extensive, the results for the same minerals should be provided for each organ and serum, because the separation into two tables (Tab. 4 with statistical significant differences) and Table 3S (non- significant data) makes it very difficult to assess the impact of diets. Like Tab.3, I suggest to separate trace elements from macroelements.
Table 4 has been re-worked according the suggestion of the Reviewer, data from Table S3 had been added and then re-organised by Macro, Trace, and Non-essential elements. Furthermore, to avoid confusion, the data relating to the serum of the rats has been combined with the data in Table S4 into a new table (Table 5) and also organised by Macro, Trace, and Non-essential elements. Table S4 and S4 have been removed from the supplementary material and the remaining tables have been re-numbered appropriately.
- Line 333-342: please check carefully the text describing the non-essential elements as I found many mistakes concerning As
The text was checked for inaccuracies and corrected. It now reads as follows:
- Lines 450-454: “As was found at concentrations above the detection limit in the soft organs of the rats (Table 4). In the brain, rats fed on the Low Ca/P + CM diet showed significantly lower As concentrations than rats fed either control diet (Control and Low Ca/P control) (p < 0.05, Table 4). In the kidney, the Low Ca/P + SM showed a significantly lower As concentration than that of the Low Ca/P control fed rats (p < 0.05, Table 4).”
- Line 353-357: also please check carefully the text describing Rb in different organs, again there are some inaccuracies
The text was checked for inaccuracies and corrected. It now reads as follows:
Line 470-475: “For Rb in the brain, liver and kidney, rats fed the control diets (Control and Low Ca/P control) showed significantly lower concentrations compared to the milk diet groups (Low Ca/P +SM, and Low Ca/P + CM) (p < 0.05, Table 4). The pattern for Rb was different in the spleen, where the Control diet fed rats showed the lowest concentration, significantly lower than the milk diet fed rats (p < 0.05, Table 4), but the Low Ca/P control diet fed rats did not show a significant difference in Rb concentrations compared with the Low Ca/P + CM feed rats (p > 0.05, Table 4).”
Reviewer 2 Report
Thank you for a well written manuscript. There are two major things I’d like addressed
- Lines 113-120. I’d like a little more information around the milk used in the feeding. I am also assuming there were no Zn boluses used in the flock or herd at the time the milk was taken. It is not clear whether a single bulk sample was used from each source or different aliquots were used for the study. From reading I believe it is the former but it would be good to be clear.
- Some development around why sheep milk has higher AL, Cs, Cu and Pb. All these elements have some negative connotations for human health and it would be good to discuss
Minor notes
- Table S2 N/A abbreviation (is this below detection limits?) needs to be subscript to the table, and the ^ consistently used within the table
Author Response
Reviewer 2:
- Lines 113-120. I’d like a little more information around the milk used in the feeding. I am also assuming there were no Zn boluses used in the flock or herd at the time the milk was taken. It is not clear whether a single bulk sample was used from each source or different aliquots were used for the study. From reading I believe it is the former but it would be good to be clear.
The milk in the trial was commercially and independently sourced. This means that the researchers are not able to provide comment on the farming practices used other than identifying that it is food grade. However, a single bulk sample was used from each source. Changes to the text have been made to highlight this as follows:
- Line 116-118: “…organic farms in the Manawatu and Hawke’s Bay regions of New Zealand, respectively. The milk used in the trial was separated into individual aliquots prior to freezing from a bulk delivery of unprocessed food grade milk, provided by each respective supplier.”
- Some development around why sheep milk has higher AL, Cs, Cu and Pb. All these elements have some negative connotations for human health and it would be good to discuss
An additional paragraph has been added to the text to cover the role of non-essential elements that are known to be potentially toxic. Additional references have been added to support this discussion and subsequent references have been re-numbered appropriately.
- Lines 195-208: “It has been previously noted in the literature that sheep milk consistently contains elevated levels of non-essential minerals in comparison to the most common ruminant milk type, cow milk [10, 12]. Of particular interest are the concentrations of Al, Cu, and Pb, because these elements are known to have a negative impact on human health. For example excessive exposure to Pb has been well linked to neurodegeneration [26]. The key mechanisms behind the elevated concentration of these elements has not been established [10, 12]. One aspect to note is the overall elevated concentration of all nutritional components in sheep milk. This means that although there can be elevated overall concentrations of individual non-essential minerals when sheep milk is compared to cow milk, the concentration of each mineral is proportionally similar between the two milk types [16]. When disproportionately higher concentrations of non-essential minerals in sheep milk (when compared to cow milk) have been noted, this is typically related to on farm variables including animal feeding patterns and contaminated milking equipment [10, 27]. In the context of the data presented in the present work (Table 3 and Table S2) the levels observed are not of concern and are within the legal limits for New Zealand and Australia [28].”
- Table S2 N/A abbreviation (is this below detection limits?) needs to be subscript to the table, and the ^ consistently used within the table
To avoid confusion the abbreviation has been changed to BDL in all tables and the table caption has been altered to reflect this. The ^ notation has also been moved to the table title to clarify the abbreviation